# Insights into mobile health application market via a content analysis of marketplace data with machine learning

Gokhan Aydin [1]*, Gokhan Silahtaroglu[2]

1 Department of Health Management, Istanbul Medipol University, Beykoz, Istanbul, Turkey, 2 Department of Management Information Systems, Istanbul Medipol University, Beykoz, Istanbul, Turkey

* gaydin@medipol.edu.tr

## Abstract

### Background

Despite the benefits offered by an abundance of health applications promoted on app marketplaces (e.g., Google Play Store), the wide adoption of mobile health and e-health apps is yet to come.

### Objective

This study aims to investigate the current landscape of smartphone apps that focus on improving and sustaining health and wellbeing. Understanding the categories that popular apps focus on and the relevant features provided to users, which lead to higher user scores and downloads will offer insights to enable higher adoption in the general populace. This study on 1,000 mobile health applications aims to shed light on the reasons why particular apps are liked and adopted while many are not.

### Methods

User-generated data (i.e. review scores) and company-generated data (i.e. app descriptions) were collected from app marketplaces and manually coded and categorized by two researchers. For analysis, Artificial Neural Networks, Random Forest and Naïve Bayes Artificial Intelligence algorithms were used.

### Results

The analysis led to features that attracted more download behavior and higher user scores. The findings suggest that apps that mention a privacy policy or provide videos in description lead to higher user scores, whereas free apps with in-app purchase possibilities, social networking and sharing features and feedback mechanisms lead to higher number of downloads. Moreover, differences in user scores and the total number of downloads are detected in distinct subcategories of mobile health apps.

**Data Availability Statement:** The dataset is hosted on OSF. Researchers can access the dataset via the following link and DOI: https://osf.io/fm5vc/.

**Funding:** GA, Istanbul Medipol University Scientific Research Projects Fund. Project Grant No: 2019/08

URL: www.medipol.edu.tr. The sponsor haven't played an active role in the study design, data collection, analysis, decision to publish, or preparation of the manuscript.

**Competing interests:** The authors have declared that no competing interests exist.

## Conclusion

This study contributes to the current knowledge of m-health application use by reviewing mobile health applications using content analysis and machine learning algorithms. The content analysis adds significant value by providing classification, keywords and factors that influence download behavior and user scores in a m-health context.

## Introduction

Maintaining public health costs and sustaining or improving public health is becoming harder throughout the world with the steady increase in median age and related diseases (e.g., diabetes, high blood pressure, etc.). This phenomenon is expected to lead to congestion in healthcare systems and higher healthcare costs. Consequently, preventive measures that can lead to a healthier life even at a later age emerge as a likely solution [1,2]. Given the recent Covid-19 pandemic, the fragility of the healthcare systems worldwide has become more evident. Within this context, the effective use of mobile technologies and devices as interventions in sustaining health and well-being come forward as a promising venue for policymakers and relevant stakeholders, which is drawing growing interest from researchers as well [3–6]. Moreover, mobile devices and apps are used more frequently due to the remote working and lockdowns that resulted from the Covid-19 pandemic, also drawing the interest of the public towards health & fitness apps [7]. However, to offer any value to the general public, mobile apps must be downloaded, installed, and actively used by the users. As of 2019, more than seven billion people (95% of the global population) live in an area covered by a mobile cellular network (GSM). Furthermore, mobile broadband subscriptions that are required for the effective use of smart mobile devices, such as smartphones and tablets, have grown by more than 20% annually over the past five years and reached 4.1 billion globally the end of 2019 [8]. Thus, the infrastructure for the mobile health (m-health) initiatives is in place in most locations. Nevertheless, among the hundreds of thousands of mobile applications on display in mobile app marketplaces, attracting attention to them is not straightforward and the success of individual apps is related to the download behavior of users. Certain apps are perceived to be more successful and effective than others, yet most are free to download. Considering similar functionalities, cost alone may not be indicative of performance or effectiveness entirely. Moreover, bearing in mind the different categories of health and wellbeing that mobile apps may focus on, different features may be influential on user choice and sentiment [9,10].

Against this backdrop, this study aims to be instrumental for policymakers, health institutions and mobile app developers by contributing to the discussion on e-health and m-health usage behavior and provide practical implications on ways to increase m-health app use. Cultural factors and ever-changing technology and applications lead to a need for continued scientific studies to identify the best practices and pave the way to wider adoption of mobile applications for sustaining and improving health. Within this context, by analyzing both user-generated (i.e. user review scores) and company generated data (app definition, description and technical data), this study aims to:

- Describe the m-healthcare characteristics of apps available on Google Play Store.

- Determine the categories that m-health apps focus on and highlight the areas neglected areas.

- Determine the app features and categories that users perceive more positively (i.e. higher user scores).

- Identify the best practices and features (e.g., keywords) through a content analysis of application descriptions.

- Test for potential relationships between data protection practices highlighted in app descriptions and download behavior/scores.

- Identifying the barriers and pain points users highlight in each mobile app subcategory.

- Provide actionable insights such as features and keywords that can be used in promoting m-health apps.

Relevant studies have mostly been carried out in western countries particularly the US and a research gap is evident in emerging countries. Adaptability of these studies' findings to emerging economies is questionable due to factors such as legislation and culture. Thus, findings from this study can be used to inform future policy development, mobile health application, planning, design and the development of m-health apps specifically in emerging economies.

This article is organized in three main sections. The first part of this paper introduces different ways mobile apps can be used for improving and sustaining health and common relevant classification methods. In this section we also provide information on mobile app features, privacy concerns and the effect of price on mobile app use in this section. The second part examines the research methodology, which is followed by the data analysis section, where the data analysis algorithms utilized along with the results are provided. Finally, we discuss the findings in the discussion section and finalize the paper by offering possible future research directions in the conclusion section.

## Background and related work

### Mobile health and health applications

The rapid adoption of smart devices in the last decade has greatly contributed to the promise of using mobile technologies for health improvement. The m-health (i.e. mHealth) term also emerged, which was defined by the World Health Organization (WHO) as: "medical and public health practice supported by mobile devices, such as mobile phones, patient monitoring devices, personal digital assistants (PDAs), and other wireless devices" [11]. Mobile devices provide good platforms for developers to design third-party apps called mobile apps, software programs that are specifically designed to run on mobile devices to improve the functionality of mobile devices. Mobile applications installed on the mobile devices can utilize hardware and sensors (i.e. accelerometers, gyroscopes, magnetometers, sensors to measure heart rate, geo sensors GPS and cameras) to obtain the desired outputs. Consequently, mobile apps provide new methods for the continuous monitoring of biological, behavioral or environmental data, health indicators, and trends related to health behavior. Mobile apps can help change attitudes and behaviors by distributing, collecting, processing and interpreting health-related information and by enabling interventions [12,13]. Therefore, various objectives may be met through mobile applications targeting a wide range of user groups. It is possible to develop applications targeting healthcare professionals, healthcare recipients and the general public. Mobile devices and applications are being used for the rapid delivery of clinical information to healthcare workers to equip rural healthcare personnel with up-to-date information in both developed and underdeveloped countries. Yet, since the applications targeting health personnel are not within the scope of this study, only the applications targeting general public and healthcare recipients are discussed from this point on. Such mobile health interventions have achieved success in varying degrees in adherence to medication and treatment outcomes [14].

The effectiveness of various cell phone application and cell phone text message interventions have been tested in clinical trials such as diabetes control [15], hypertension control [16], and adherence to medication [17]. In cases where treatment is complex, such as with cancer patients, mobile applications are also used to improve health literacy in order to improve compliance [18]. Similarly, systematic reviews on the use of various smartphone apps and exercise platforms to improve diet, physical activity and sedentary behavior, and the effectiveness of related interventions have shown positive yet modest effects [12,19,20]. Mobile apps were also found to be influential in improving emotional and mental health. In a relevant study, it was observed that self-monitoring of personal mood with a mobile application reduces symptoms related to depression and anxiety and improves emotional well-being [21]. Consequently, mobile applications may be used as preventive medical tools in a multitude of ways [2,10,14].

Several studies have been conducted to analyze the features of mobile health software on app marketplaces, yet most of them have focused on a particular application area. Correspondingly, m-health app use literature is diversified and numerous subcategories exist. Depending on the use case mobile apps may be used as information sources, journals and personal digital assistants for quitting smoking, healthy eating, reducing calorie intake, increasing physical activity levels, communicating with the health system, improving adherence to treatments (e.g., on-time drug intake), medical monitoring and more [9,10,20,22–25]. Given the variety of apps available to the general public, only a limited number of studies analyzed all available categories [e.g., 4], creating a research gap. Such ambitious studies called for solid frameworks to categorize the available apps and utilized several popular health education, planning and promotion models. One such framework is the PRECEDE-PROCEED model (PPM) of health planning and health education, another is Health Education Curriculum Analysis Tool (HECAT) health education content classification [4,26]. Additionally, disease management models such as WHO Global Burden of Disease [27] have been utilized in relevant studies. The PPM is a widely applied ecological approach to the planning and promotion of health interventions [28]. The PPM has been applied to the m-health applications context by adopting the tripartite structure of predisposing, enabling, and reinforcing factors [4,26]. Within this framework, predisposing factors such as mobile applications try to influence attitudes by providing information and increasing awareness of conditions and or health outcomes, as well as changing beliefs and attitudes to establish confidence among users so that they can change their behavior to avoid adverse outcomes. Enabling factors/ mobile applications on the other hand, aim to change behavior and formed habits by providing an opportunity to learn a new skill and to follow up on the progress of a subject (e.g., applications that allow daily/monthly recordings and follow-ups of running/cycling times and duration while doing sports). Lastly, reinforcing factors/applications aim to encourage certain behaviors that will help in improving and sustaining health through different reward and feedback systems provided to users [4,29]. For instance, apps with auto-sharing to social media sites such as Facebook, or apps that provide ways to communicate and get feedback from an online coach are considered in this category.

Another framework used in the literature for categorizing m-health apps, which is also used in the present study, is HECAT [4] by the Centers for Disease Control and Prevention [30]. This framework focuses on health education curricula provided to students. The categories considered within the HECAT health education content classification are as follows: Alcohol and Other Drugs, Healthy Eating, Mental and Emotional Health, Personal Health and Wellness, Physical Activity, Safety, Sexual Health, Tobacco, Violence Prevention, Comprehensive Health Education.

## Factors affecting m-health application use

**Mobile application features.**   It is obvious that there are certain macro- level barriers to the use of m-health applications such as low technology literacy, income, limited access to mobile devices, and lack of infrastructure. Yet, certain other barriers are perceived barriers and may be influenced by mobile app developers and sponsors. The findings obtained in a large-scale study in the US have indicated that a significant part of the population does not use healthcare applications due to hidden or visible costs, high data entry burden, complex systems, and data security concerns [9]. The need to understand and address user concerns is critical to ensure wide use of these applications. Perceptions of functionality, performance, trustworthiness, ease of use (e.g., interface, time required to learn etc.), and certain privacy concerns that are influential in usage [31] may be overcome by application developers through the careful design of mobile apps and communication in mobile marketplaces [32]. For instance, it has been shown that informational content, organizational attributes, technology-related features, and user control factors influence the trustworthiness of m-health applications [33]. Moreover, the mobile health app features are a research area that attracted the interest of researchers due to its significance in usage behavior [34]. Considering that these factors are commonly assessed by users while browsing app stores without experiencing the app itself, the proper use of available tools such as app descriptions is vital to success.

**Pricing.**   Pricing and costs associated with using the app have been considered in the literature as a significant factor that hinders the wider adoption of mobile devices and apps. Several studies indicated that the costs associated with mobile app use are a barrier to adoption [9,10,35,36]. Moreover, increased app prices are shown to decrease app sales [37], yet most apps are offered free of charge by developers to users in several contexts [e.g., 38]. The revenue is generated from advertising displayed to users, in-app purchases for premium features, higher convenience, etc. The latter model is also called freemium model, which has been tested and tried in software and mobile application contexts. Thus, offering basic functionality free of charge, while offering in-app purchase options for higher functionality is a viable model that is also used in a health apps context [39,40].

**Privacy, information disclosure and relevant legislations.**   Information privacy is a delicate topic in m-health apps context due to the personal and sensitive nature of the information gathered from the users [41]. Personal information is used to establish functionality and provide value to users but also to determine the content of the advertisements to be displayed to users in the case of free applications. Advertisement content such as for medical products may or may not be certified and their effectiveness or safety is questionable given the lack of effective control mechanisms. Keeping the personal information that is collected through apps secure and being transparent are vital issues for mobile application developers and sponsors [10,42]. User privacy concerns should be addressed and compliance with legal regulations and ethical concerns should be met [9,41,43]. Although attempts have been made to establish standards for mobile applications being released in the field of health, unfortunately, there is no globally accepted framework. The American Food and Drug Administration (FDA) emphasized the necessity of maintaining certain technical standards and data protection in the US and tried to set certain standards in the mobile application ecosystem. The FDA aims to establish standards within this framework in private institutions to develop applications that offer "reliable content, protect information privacy and security, and work as promised" [44]. Regardless of the efforts, control mechanisms are not sufficient in practice and health apps that are estimated to be in the hundreds of thousands, carry many security concerns as keeping the information secure is not straightforward [45,46]. Studies on mobile apps suggest that apps targeting patients could perform better, particularly regarding privacy and security issues [e.g.,

47]. Despite its significance, privacy concerns have not been addressed properly in most m-health apps as studies suggest. For instance, in a review study on health applications in mobile application markets, only 30% of the 600 mobile applications evaluated were found to have a privacy policy. Moreover, approximately two-thirds of the available privacy policies were generic [48]. These findings indicate that there are deficiencies in providing the necessary transparency regarding the information collected and its security. This may be among the reasons why users prefer not to download certain apps.

## Research methodology

To attain the research objectives, a cross-sectional study of m-health apps was performed to characterize and classify apps. Publicly available data on free and paid apps provided by app developers and users on the Google Play app store were collected, coded, classified and analyzed as detailed throughout this section. The Istanbul Medipol University Ethical Committee of Non-invasive Clinical Trials specifically approved the present study (document no: 10840098–604.01.01-E.8356).

### Selection, data collection and screening

Mobile apps are selected by browsing the health and wellness category of Google Play app store and carrying out searches using 'health' and 'wellness' as keywords. The data on apps was collected in the second half of January 2020 via an application programming interface (API) in Python programming environment. Data on 520 free and 520 paid mobile applications classified under the health and wellness category were retrieved through the API. The app selection and exclusion process can be seen in Fig 1.

### Data coding and classification and validation

In the data coding stage, 30 duplicates and 36 apps belonging to other categories or having different purposes (e.g., games, gym membership apps etc.) were left out of further analysis. App description data were cleaned with text mining tools. Firstly, all descriptions were translated

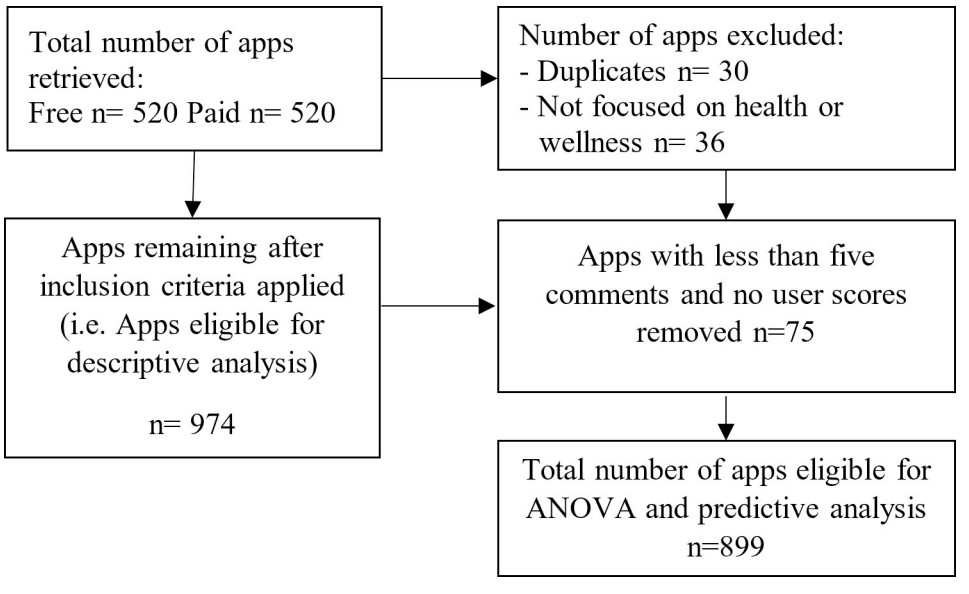

**Fig 1. App selection process.**

into a single language i.e. Turkish as not all were in the same language. After that, texts have been tokenized and stemmed (i.e. converted to its base form). Following this process goes, going, went or gone were converted to their base form, 'go'. Then, stop words (i.e. the most common words in a language) and punctuation filtering were applied to the data. In this stage words such as 'a', 'an', 'the', 'but', 'only' was filtered. Additionally, punctuation marks such as '?', ';' and bullet points were also removed from the dataset. After this process, all text data have been vectorized, in other words converted to binary dummy variables. Since this creates a large dataset that is hard to administer, frequency elimination was applied. Classical ways of calculating the weighting of words are term-frequency and Document Frequency (DF). DF is the number of documents in which the term appears. One may think of the definite article 'the', which occurs more than any other lexical words in an English text. However, it does not mean that 'the' is more informative than other words are. For this purpose, the Inverse Document Frequency (IDF) method was used. This method applies a technique considering all reviews and the frequency of each word in each document inversely. In this way, if a word occurs too many times in all of the documents, its frequency value drops. The description data which was vectorized were merged with other variables. In the Eq (1), fr_t,d is the frequency of each term in the documents, while 'd' represents the number of documents in which the term occurs. In our case d is the number of reviews.

$$TF_{t,d} = \frac{fr_{t,d}}{\sqrt{\sum_{t=1}^{n} fr_{t,d^2}}} \tag{1}$$

Inverse document frequency (IDF) is a measure which aims to deliver how much information the term gives. IDF measure does not count on whether the term occurs frequently or rarely. It takes one or more documents in the corpus. N is the total number of documents in the corpus and nt is the number of documents in which the term appears. If the word does not occur in the document nt = 0, this leads to a division-by-zero case. In order to avoid this problem, it is smoothed with 1 + nt.

$$TF - IDF = TF_{t,d} \cdot \frac{N}{nt + 1}) \tag{2}$$

Regarding the manual data classification, two distinct frameworks were used to categorize the mobile applications. First, the PPM [28] was utilized to categorize mobile applications with regard to how they promote health (i.e. their major aims and how they provide value to users). Within this framework, mobile applications were classified as predisposing, reinforcing, or enabling. This model has been widely used in health education planning, promotion, diagnosis and evaluation activities [49]. Similar to the methodology followed by West et al. [4] in a related study, the applications have been grouped with regard to the PPM framework in this study as well. Consequently, the coders coded the applications as predisposing, enabling or reinforcing. Within this framework m-health apps were coded as predisposing if they provided health information and statistics related to influencing attitudes, knowledge, awareness, beliefs and values or if they aimed to increase the confidence, motivation or self- efficacy of users (e.g., an application that provides smoking and cancer related statistics; ways to avoid adverse health outcomes). If the apps were used for tracking or recording status or progress (e.g., weight or calorie counting, geo-locating running/biking activities), or they facilitated behavior by teaching a skill (e.g., an app showing images or videos on proper posture) they were coded as enabling. Enabling apps are commonly used at the same time as the desired behavior. The apps were coded as reinforcing if they interfaced with a social networking site (e.g., apps with an automatic upload to Facebook), provided encouragement from trainers/coaches (e.g., an

app that featured easy communication with a trainer), and included an evaluation based upon the user's self-monitoring (e.g., an app that provided automated notifications). In the event that an app was both considered enabling or reinforcing, the reinforcing category was coded. If an app can both be considered predisposing or enabling the enabling category was coded. Examples of applications that were coded as predisposing are Health Articles, Info & Motivation–Lets Healthify, Ketogenic Diet and EO Guide. Applications that were coded as reinforcing are Drink water reminder, Soccer Training and Quit Smoking Tracker GOLD; coded as enabling are Period Tracker Mia, YAZIO Calorie Counter, Nutrition Diary & Diet Plan, and Calorie Counter–MyFitnessPal.

The second classification method used is HECAT health education content classification by the Centers for Disease Control and Prevention [30]. The categories of coding within the HECAT health education content classification are: AOD: Alcohol and Other Drugs, HE: Healthy Eating, MEH: Mental and Emotional Health, PHW: Personal Health and Wellness, PA: Physical Activity, S: Safety, SH: Sexual Health, T: Tobacco, V: Violence Prevention, CHE: Comprehensive Health Education.

Two research assistants working at a management information systems department with a focus on healthcare, who provided informed consent to participate in this study, were trained by the authors to collect and code application data into the aforementioned categories based on the two frameworks discussed. Two training sessions were held over the course of data coding by two authors and two research assistants, one before coding and another after the coding of 60 apps in a pilot study. The manual coding detailed in this section helped in obtaining further variables (i.e. the PPM and HECAT categories, app sponsor type, privacy policy availability) in addition to the information retrieved from Google Play Store. Table 1 presents all of the metrics and qualitative parameters that were extracted in this manner and fed into the machine learning models. These predictor variables were used in the analysis to predict the total number of downloads. Supervised learning algorithms need a class variable for training using the available data. In the current study, 'the total number of downloads' was chosen as the class variable parallel to the main aim of this study, which is to discover the ways to improve the use of m-health apps.

Inter-coder reliability was computed on 120 apps (approximately 12% of the total dataset) after the apps were coded. The degree of agreement between the two coders was calculated and the overall 87% agreement figure led to the conclusion that there was no significant inter-coder issue.

## Data analysis and results

Following the re-coding process, the frequencies of each variable and the results of the classifications are provided in Table 2.

As a next step, the means of different categories were compared via ANOVA analysis on SPSS software package to reveal the factors that are influential in review scores. User scores variable was set as the dependent variable whereas the app sponsor, PPM categories, HECAT categories, content ratings, price groups, being an editors' choice, having a privacy policy, having a video in the description, in-app purchases, the number of days since last update and interactive elements were set as independent variables. 75 apps out of 974 did not have any published review scores, thus were left out of this analysis. The results of the analysis are detailed in Table 3. According to the results, the PPM category, content rating, price group, the days since last update and the interactive elements were not significantly different in terms of user scores and F values are not provided for these variables in an effort to save space.

As a further step of the analysis, three machine learning algorithms were used to predict the number of downloads. The overall workflow of the machine learning analysis is depicted in Fig 2.

**Table 1. Code sheet: Metrics and parameters.**

| Variable | Values | Description |
|---|---|---|
| Video | Yes–No | Whether there is a video provided by the developer/sponsor in the app description or not. |
| Description | Free text | Text provided by app developer to describe the app on marketplace. |
| Editor's Choice | Yes–No | Editor's Choice badge given to app or not. |
| Free | Yes–No | Free or paid app. |
| Price | 0 or a double value in US dollars. | Price to be paid to run the app; recoded into 5 categories (see Table 2). |
| Days since last update | Integer between 11–2371 | Days since the app was last updated as of the data collection date. Recoded into 5 categories (see Table 2). |
| In App Purchase | Yes–No | In-app purchases provided or not |
| Size | double value in Bytes | Size of the downloaded app package |
| Required Android Version | Text | Name of the version such as 4.4 or up, 5.0 or up. |
| Content Rating | Everyone or Teen | Everyone or Teen with warning such as 'use of alcohol, gambling, Language' |
| Interactive Elements | None, Digital Purchases, Users Interact, Shares Location, Shares Info. | Info on interactivity provided in the app such as purchases, sharing and user interactions. |
| Score | Double value between 1–5 | Score provided by users to the app. (min.5 reviews required for score). |
| Number of Reviews | Integer between 0–2,125,979 | Total number of reviews users have provided on an app. |
| Installs | 10+, 50+, 100+, 500+, 1K+, 5K+, 10K+, 50K+, 100K+, 500K+, 1M+, 5M+, 10M+ | Number of times app is installed (categorized by Google app market) |
| App Sponsor/ Origin | Government, large corporation, SME-Developer, individual | The main sponsor of the app |
| Privacy Policy | Yes–No | Whether privacy policy is mentioned in description or not. |
| HECAT App Category | AOD: Alcohol & Other Drugs, HE: Healthy Eating, MEH: Mental & Emotional Health, PHW: Personal Health & Wellness, PA: Physical Activity, S: Safety, SH: Sexual Health, T: Tobacco, V: Violence Prevention, CHE: Comprehensive Health Education | A new category 'Maternal and Infant Health'was amended to the existing HECAT classification given that there are several apps aiming at new mothers and parents. AOD and T categories are joined as TAOD to carry out certain analysis due to the low number of apps in these categories. |
| PPM App Category | Predisposing, Enabling, Reinforcing | Details on PPM classification are provided in Section 2.1. |

A decision tree model was used to extract hidden patterns behind the number of downloads. Decision trees create IF-ELSE-RULES which can be interpreted by humans and can also be implemented in third -party applications. The most frequently used decision tree algorithms are Gini and Entropy (gain ratio) based algorithms. However, it is well known that they are weak learners. They are easily affected by dataset variations and outliers. In order to circumvent this overlearning or overfitting problem, ensemble and random forest decision tree (RFDT) models have been developed. They yield successful results on many data bases. In this study, the decision tree model we have employed is the random forest model. As the name suggests, the random forest model creates more than one tree. Trees are created with randomly selected variables from the dataset by using different algorithms [50]. The total number of the trees in the forest is determined by the data scientist. The root of the tree is essential for a decision tree model. It is considered to be the most important parameter to explain the target class variable. Thus, in a forest it is critical to know how many times each variable was selected as the algorithm to be included in the root of the trees. Traditionally, random forests may use GINI or Entropy based decision tree algorithms. In this study, GINI based random forest algorithm, which employs probability concept as indicated in Eq (3), was used. In the equation, p

**Table 2. Frequencies of apps and app classification(s).**

| Variable | # of apps | % of total | Variable | # of apps | % of total |
|---|---|---|---|---|---|
| **Sponsor/Origin** | | | **Video in Description** | | |
| Corporation | 175 | 18.00% | Yes | 193 | 19.80% |
| Government | 10 | 1.00% | No | 781 | 80.20% |
| Individual | 132 | 13.60% | **Days since last update** | | |
| SME-Developer | 657 | 67.50% | 1-60days | 186 | 20.50% |
| **PPM Categories** | | | 61-120days | 151 | 16.60% |
| Enabling | 669 | 68.70% | 121-240days | 171 | 18.80% |
| Predisposing | 159 | 16.30% | 240-365days | 128 | 14.10% |
| Reinforcing | 146 | 15.00% | 365+days | 273 | 30.00% |
| **HECAT Categories** | | | **Privacy Policy** | | |
| Maternal&Infant Health | 32 | 3.00% | Yes | 81 | 8.30% |
| CHE | 43 | 4.10% | No | 893 | 91.70% |
| HE | 149 | 14.10% | **Price (USD)** | | |
| MEH | 150 | 14.20% | Free | 500 | 50.00% |
| PA | 368 | 34.80% | 0.99–1.50 | 107 | 11.00% |
| PHW | 247 | 23.40% | 1.51–3.00 | 175 | 18.00% |
| SH | 44 | 4.20% | 3.01–5.00 | 103 | 10.80% |
| TAOD | 23 | 2.20% | 5.01+ | 89 | 9.20% |
| **Version Required** | | | **Installs** | | |
| 1.6 and up | 5 | 0.50% | 10+ | 54 | 5.60% |
| 2.1 and up | 12 | 1.20% | 100+ | 80 | 8.20% |
| 2.2 and up | 46 | 4.70% | 500+ | 63 | 6.50% |
| 2.3 and up | 29 | 3.00% | 1,000+ | 174 | 17.90% |
| 2.3.3 and up | 11 | 1.10% | 5,000+ | 75 | 7.70% |
| 3.0 and up | 10 | 1.00% | 10,000+ | 120 | 12.30% |
| 4.0 and up | 94 | 9.70% | 50,000+ | 61 | 6.30% |
| 4.0.3 and up | 122 | 12.50% | 100,000+ | 133 | 13.70% |
| 4.1 and up | 249 | 25.60% | 500,000+ | 41 | 4.20% |
| 4.2 and up | 60 | 6.20% | 1,000000+ | 103 | 10.60% |
| 4.3 and up | 24 | 2.50% | 5,000,000+ | 19 | 2.00% |
| 4.4 and up | 122 | 12.50% | 10,000,000+ | 51 | 5.20% |
| 5.0 and up | 94 | 9.60% | **Editors' Choice** | | |
| 6.0 and up | 20 | 2.00% | No | 942 | 96.70% |
| Varies with device | 76 | 7.80% | Yes | 32 | 3.30% |
| **Average Rating** | | | **Content Rating** | | |
| 1.5–3.0 | 50 | 5.70% | Everyone | 928 | 95.30% |
| 3.1–4.0 | 214 | 24.60% | Teen | 34 | 3.50% |
| 4.1–4.5 | 332 | 38.10% | Mature (17+) | 12 | 1.20% |
| 4.6–5.0 | 275 | 31.60% | | | |
| *Total* | *974* | *100%* | *Total* | *974* | *100%* |

is the probability of a variable being in a field of the data set.

$$G(split) = \sum_{i=1}^{n} \frac{n_1}{n} \left(1 - \sum_{j=1}^{n} p^2\right) \tag{3}$$

The second machine learning model we used is Artificial Neural Networks (ANN). ANN are composed of layers, which are interconnected via weights as visualized in Fig 3. The

**Table 3. F-test compare means results.**

| Variable | Mean | N | Std. Dev. | Min. | Max. | F-value | Sig. |
|---|---|---|---|---|---|---|---|
| **Video in Description** | | | | | | | |
| No | 4.168 | 702 | 0.6125 | 1.5 | 5.0 | 5.124 | .024** |
| Yes | 4.275 | 197 | 0.4618 | 2.2 | 5.0 | | |
| Total | 4.192 | 899 | 0.5842 | 1.5 | 5.0 | | |
| **HECAT** | | | | | | | |
| Maternal & Infant Health | 4.438 | 26 | 0.4900 | 3.0 | 4.9 | 5.367 | .000*** |
| CHE | 4.100 | 34 | 0.5836 | 2.7 | 4.9 | | |
| HE | 4.035 | 123 | 0.5989 | 2.2 | 5.0 | | |
| MEH | 4.283 | 125 | 0.5059 | 2.3 | 5.0 | | |
| PA | 4.275 | 335 | 0.5474 | 1.5 | 5.0 | | |
| PHW | 4.053 | 199 | 0.6405 | 2.0 | 5.0 | | |
| SH | 4.220 | 35 | 0.5825 | 2.2 | 5.0 | | |
| TAOD | 4.336 | 22 | 0.6314 | 2.4 | 4.9 | | |
| Total | 4.192 | 899 | 0.5842 | 1.5 | 5.0 | | |
| **Sponsor/Origin** | | | | | | | |
| Corporation | 4.119 | 159 | 0.6397 | 1.5 | 5.0 | 3.930 | .008*** |
| Government | 3.680 | 10 | 0.5574 | 2.7 | 4.4 | | |
| Individual | 4.246 | 121 | 0.5271 | 2.6 | 5.0 | | |
| SME Developer | 4.208 | 609 | 0.5760 | 1.8 | 5.0 | | |
| Total | 4.192 | 899 | 0.5842 | 1.5 | 5.0 | | |
| **Privacy Policy** | | | | | | | |
| No | 4.181 | 819 | 0.5833 | 1.5 | 5.0 | 3.103 | .078* |
| Yes | 4.301 | 80 | 0.5862 | 2.2 | 4.9 | | |
| Total | 4.192 | 899 | 0.5842 | 1.5 | 5.0 | | |
| **Editors' Choice** | | | | | | | |
| False | 4.179 | 867 | 0.5884 | 1.5 | 5.0 | 11.774 | .001*** |
| True | 4.538 | 32 | 0.2938 | 3.8 | 4.8 | | |
| Total | 4.192 | 899 | 0.5842 | 1.5 | 5.0 | | |
| **In-App Purchase** | | | | | | | |
| False | 4.133 | 651 | 0.6132 | 1.5 | 5.0 | 21.567 | .000*** |
| True | 4.344 | 248 | 0.4681 | 2.2 | 4.9 | | |
| Total | 4.192 | 899 | 0.5842 | 1.5 | 5.0 | | |
| **Install Categories** | | | | | | | |
| 1,000,000+ | 4.418 | 173 | 0.4480 | 2.0 | 4.9 | 12.882 | .000*** |
| 1,000+ | 4.109 | 172 | 0.5961 | 2.3 | 5.0 | | |
| 10–1000 | 4.194 | 125 | 0.6303 | 2.2 | 5.0 | | |
| 5,000–50,000 | 4.013 | 195 | 0.6759 | 1.5 | 5.0 | | |
| 50000–1,000,000+ | 4.232 | 234 | 0.4931 | 2.5 | 4.9 | | |

\* significant at 0.10< level

\*\* significant at 0.05< level

\*\*\* significant at 0.01< level.

simplest ANN has one input layer (variables used for learning), a hidden layer and an output layer (class variables). Hidden layers contain neurons that hold activation functions. Each neuron makes a decision about how strongly it should activate. Each neuron is also fed with weights coming from previous layers. A backpropagation method is applied to adjust weights

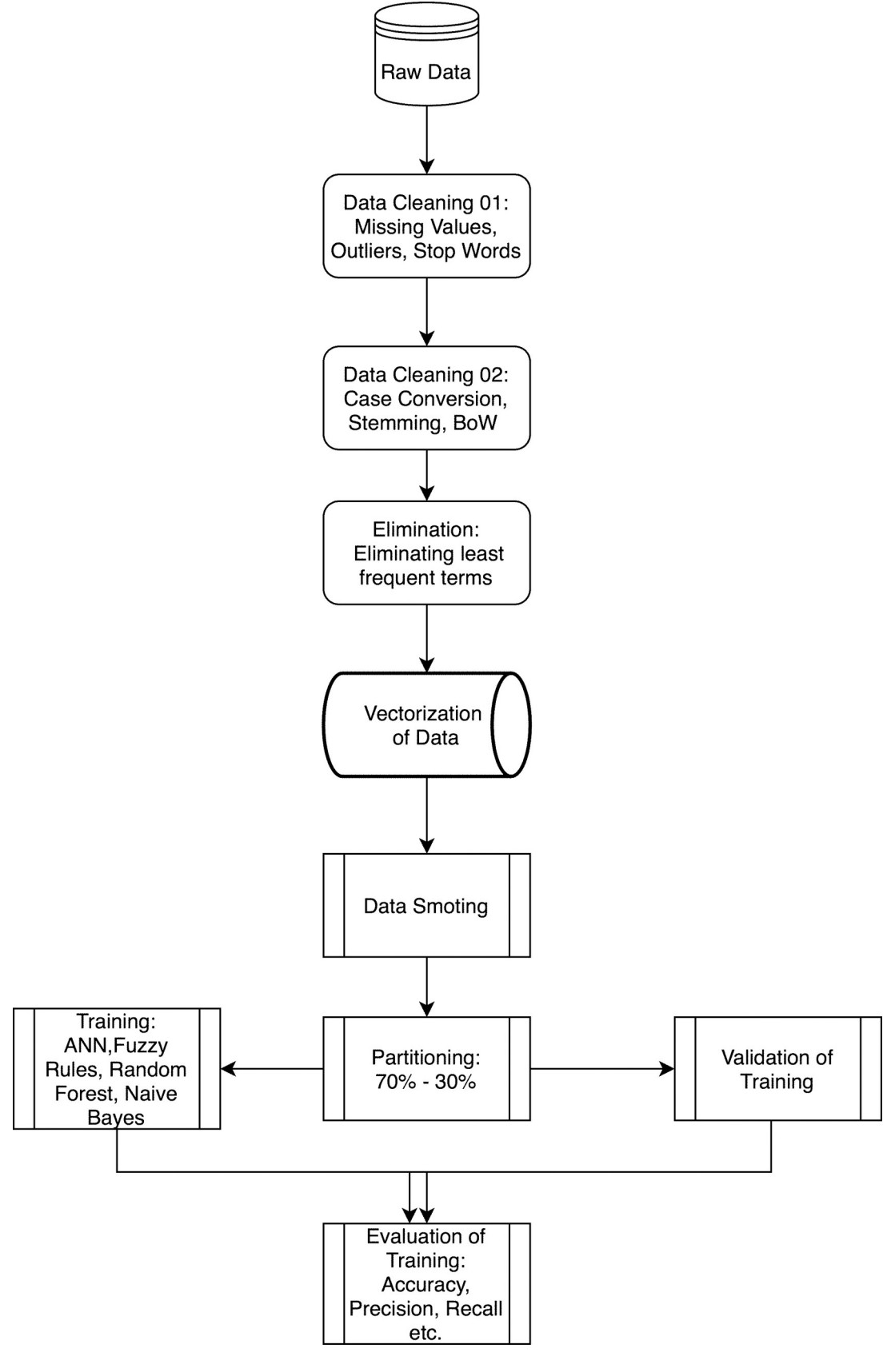

**Fig 2. Overall workflow.**

with minimum prediction error. ANN generates weight matrixes as learning outputs [51]. Neural networks need to be fined-tuned with certain hyperparameters.

Thirdly, Naive Bayes (NB) machine learning models, which are one of the most popular probabilistic classifiers in data science, were used. NB's are based on the Bayes Theorem of conditional probabilities. They tolerate noise and outliers well; their training time is relatively short, and they need few hyperparameters for fine-tuning. In the Eq (4) P(x) is probability of x in the document (data set), P(c) is the probability of the labeled class in the data set and finally P(x/c) is the probability of variable x in a given class.

$$P_{(c/x)} = \frac{P_{(x/c)} P_{(c)}}{P_{(x)}} \tag{4}$$

The KNIME analytics platform was used to generate all machine learning models and to apply algorithms. For all the algorithms applied, 30% of the data was used for validation, and 70% was used for training. Stratified sampling method was used for partitioning dataset as learning and validation parts. To overcome minority class problem, Synthetic Minority Over-sampling Technique (SMOTE) was employed. Hyperparameter selection for the algorithms utilized is as follows:

- The quality measure was chosen as Gini index for random forest decision tree model. The number of levels was set at 10 and the minimum node size was 3. The n-estimator was chosen as 100. A 5-fold sampling (without replacement) was done along with stratified sampling.

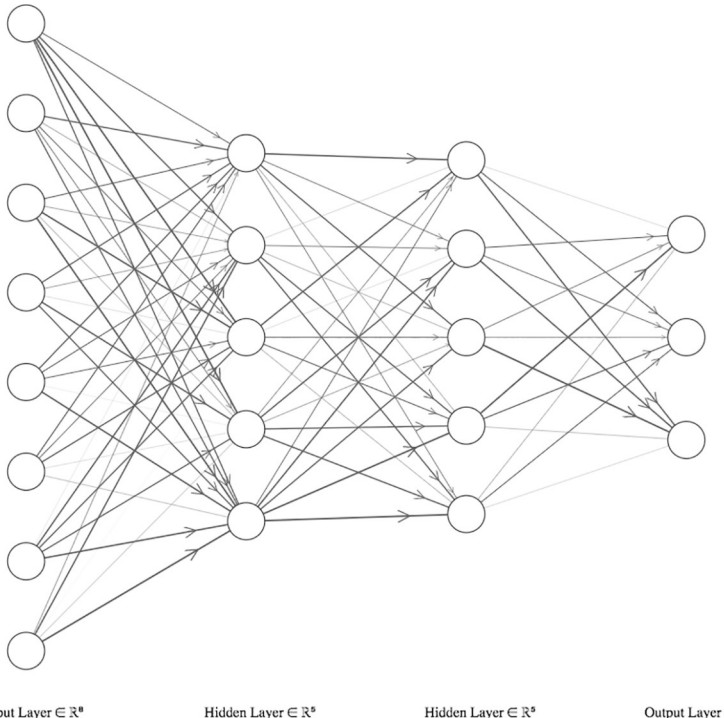

Input Layer $\in \mathbb{R}^8$ Hidden Layer $\in \mathbb{R}^5$ Hidden Layer $\in \mathbb{R}^5$ Output Layer $\in \mathbb{R}^3$

**Fig 3. Artificial neural networks.**

**Table 4. Model prediction performance.**

| Algorithm | Precision | Sensitivity | Specifity | F-Measure | Accuracy | Cohen's Kappa |
|---|---|---|---|---|---|---|
| ANN | 0.719 | 0.700 | 0.908 | 0.704 | 0.738 | 0.627 |
| NB | 0.670 | 0.663 | 0.888 | 0.662 | 0.660 | 0.550 |
| RFDT | 0.724 | 0.722 | 0.909 | 0.720 | 0.724 | 0.632 |

- In ANN, sigmoid activation function was preferred, and z-score normalization was applied to the dataset to speed up the training. Stochastic depth and early stopping were to prevent any possible overfitting. The best performing model for ANN was achieved with two layers and 10 nodes in each layer after several trials.

We calculated the accuracy, precision, sensitivity, specifity, Cohen's Kappa, and F- values using true positive, true negative, false positive, and false negative cases to evaluate the learning quality and performance of the algorithms as provided in Table 4 and Fig 4.

The influences of different factors on download behavior were assessed using machine learning algorithms. When the results of these analyses are assessed, the best learner emerges as the ANN, which can be seen in Table 4. The output of the ANN analysis is not directly discussable, yet the sensitivity of each input variable can be calculated and interpreted. Thus, we calculated the sensitivity of input variables using the weights between inputs and the first layer of the ANN. The weights were then normalized by dividing the weight of each variable by the grand total of the weights. The sensitivity analysis of the Top-100 parameters (consolidated and filtered) led to the following influential keywords:

- **Keywords:** part, young, band, weight, period, breath, pedometer, guts, treatment, calculate, education, smoking, use, outside, activity, timer, running, water, educate, Bluetooth, development, sutra, chest, glycemic, step, cycle, run, dance, points, plans, use, photo, music, minute, notification, progress

Moreover, a second run of the ANN by excluding the description text data has led to the findings provided in Table 5. Please note that random forest and naive Bayes algorithms have not been used further since they yielded relatively weaker accuracy and precision.

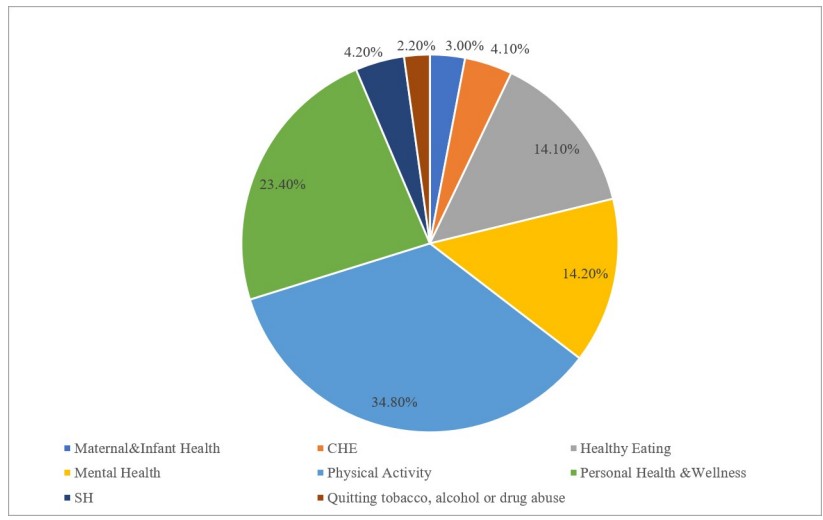

**Fig 4. HECAT application categories.**

Table 5. ANN sensitivity (excluding description text data).

| Category | Sensitivity |
|---|---|
| Required Android Version | 21.75% |
| Interactive elements | 16.42% |
| HECAT category | 14.42% |
| Content Rating | 12.95% |
| Free | 7.63% |
| Sponsor | 4.95% |
| User Score | 4.60% |
| In-app Purchases | 4.17% |
| Privacy | 3.97% |
| PPM | 2.73% |
| Days since last update | 2.21% |
| Editor's choice | 2.19% |
| Price | 1.29% |
| Video | 0.72% |

The top five factors that are related to the total number of downloads is the 'Required Android Version', 'Interactive elements', 'HECAT category', 'Content Rating' and whether the app is 'free' or not. Lastly, the second-best performing algorithm RFDT's tree attribute selection results highlight the role of the following parameters on download behavior:

- **Variables:** Sponsor, HECAT, Editor 's Choice, Android version, Free.

- **Keywords:** Light, Routine, Follow, Information, Morning, Food, Exercise, Step recorder, Ovulation period, Develop, Relax, Friends, Heart.

These findings are discussed in the following section in detail.

## Discussion

The most common m-health app category among the sample was Physical Activity (34.8%) as indicated in Table 2 and as visualized in Fig 5. This was followed by Personal Health and Wellness (23.4%), Mental Health (14.2%), and Healthy Eating (14.1%). Apps that target quitting tobacco, alcohol use and/or drug abuse attracted the least attention (2.2%) from developers in the sample. It is evident that certain categories are represented and promoted at a higher frequency that other categories, hinting at that there may be areas where competition/supply is less intensive.

When the user scores provided in Table 3 were assessed, it became evident that there were significant differences in user scores between HECAT app categories. Moreover, RFDT analysis also indicated that the HECAT category played an important role in download behavior (i.e. app popularity). The lowest average score was given to Healthy Eating apps by users among all categories (4.04). Despite the presence of a relatively high number of apps focusing on this category, users were not entirely content with Healthy Eating apps they have downloaded. This indicates that there is room for improvement in this subcategory, which may benefit app sponsors may when launching new apps or improving existing ones. Another common app category (23.4% of the total sample) that received relatively low scores (4.05) was the Personal Health and Wellness category that encompasses a larger variety of apps than other categories. The low scores indicated that despite the high number of apps and variety, user satisfaction is yet to be established. On the other hand, mobile applications on 'Maternal

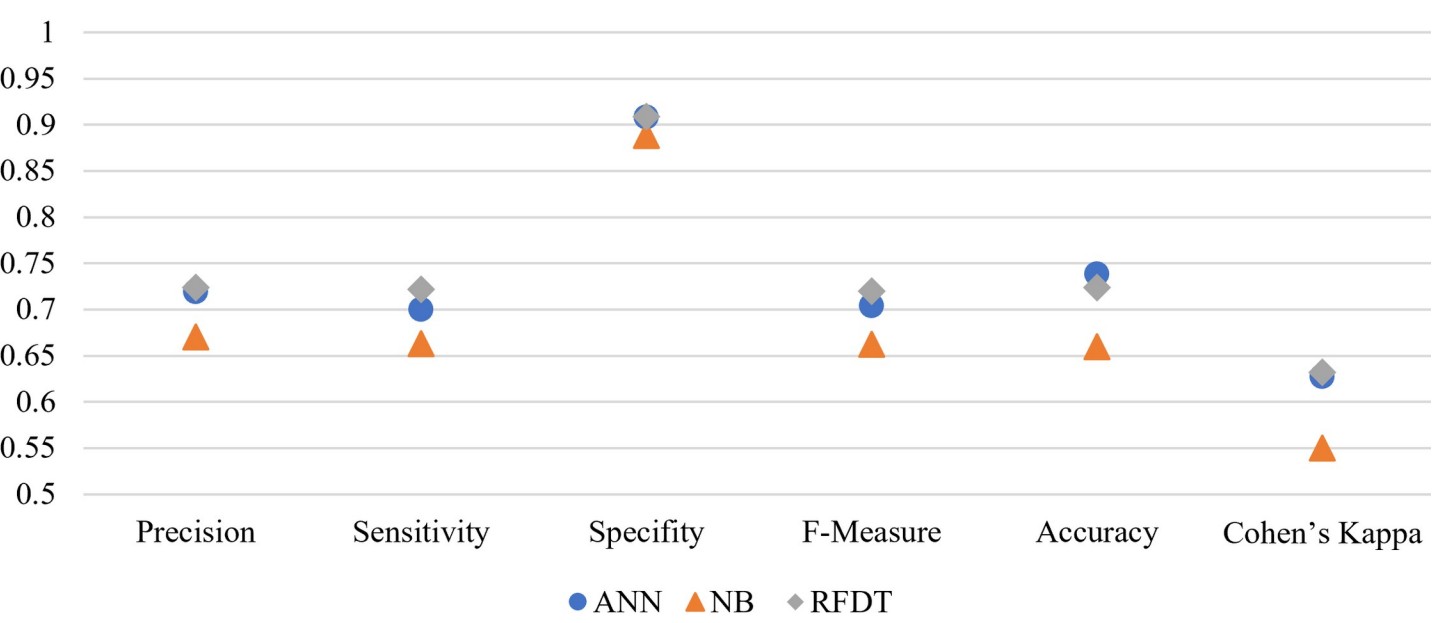

**Fig 5. Algorithm prediction performance results.**

and Infant Health' received the highest average score (4.44) among the total sample. This specific category targeting new moms and parents received the most positive scores, indicating that competition may be challenging in this category where the users are already particularly satisfied with the available apps. Surprisingly, apps for quitting smoking, alcohol and other drugs have received low interest from developers (~2% of the total) despite the high user scores (4.34) and emergence of 'smoking' as a significant keyword related to the number of downloads. Given the high percentage of smokers (28% of the total population) in Europe [52] and the risks associated with alcohol use [53], more apps may be promoted to users. Moreover, there were no apps promoted by governmental institutions in the TAOD category among the sample. More resources may be allocated by governmental institutions on this category to disseminate apps to more users.

As discussed in the background section, pricing and costs associated with using the app are also influential in download behavior and have been considered as three distinct variables in this study: whether the app is free to use, the actual price to be paid, and whether in-app payment are offered to users or not. The first two factors were shown not to be influential on user scores, yet the 'free' variable proved to be influential on the total number of downloads. It is obvious that if an application is free more users will download it. The cost(s) associated with m-health app use have appeared as a barrier to adoption in several academic studies as well [9,10,36]. Thus, our finding signifying the free apps attracting more downloads is in accordance with the literature. Moreover, in-app purchases emerged as a notable factor related to user review scores. The apps offering in- app purchases received better user scores, which is in agreement with the results of a study by Biviji et al. [54]. It is evident that the freemium business model, also termed as in-app purchase strategy [39], in which basic functionality is offered freely while in-app purchase options are used for accessing extended features, works well in promoting health apps. This has been evidenced in the literature as well [40].

Another significant factor is the 'Editor's Choice', which was found to be influential in both download behavior and user scores. If the app had an editor's choice badge, both the user scores and the total number of downloads were observed to be higher. It should be noted that higher user scores and downloads may also lead to an 'Editor's choice' badge so there is no clear causality in this relationship. A similar study on app marketplaces indicated that a comparable factor, the 'best-seller rank' affects consumers choice, which supports our finding [55]. Nonetheless, this is not a feature that can directly be controlled by m-health app sponsors or developers, and therefore, it does not lead to significant applicable insights.

A further noteworthy factor, which is associated with user scores is the presence/absence of a privacy policy regarding the use of the app. The apps that provide information on privacy policies in the description section have received higher user scores than apps without such information. However, only less than 9% of the mobile apps analyzed in the study had any mention of privacy policy within their descriptions. A similar finding was obtained in a review study on mobile health apps by Sunyaev et al. [48], where only 30% of the 600 apps analyzed were found to have a privacy policy. It is evident that app developers have not improved their stance on privacy policies in the last six years since that study. A need for further action by app developers, sponsors and policymakers is required to increase transparency regarding the privacy of personal health information collected by apps. These initiatives by app developers regarding privacy policy development and use may lead to higher user satisfaction as evidenced by higher user scores in the present study.

Another significant element in the ANOVA analysis that had a positive influence on user scores is 'Videos'. This variable indicates whether there is a video provided by the app developer in the app description or not. Mobile apps with videos, which commonly show how the app works, received higher user scores than apps without videos. Interestingly, only 20% of the apps analyzed had videos provided in the description section. This leads to a practical implication that can benefit app sponsors/developers without spending considerable resources. Relevant descriptive app videos can be prepared through the use of common video maker and editor software that utilize screen captures and images, which can then be provided on marketplace app description pages.

Taking the keywords related to the app features into consideration, the ones related to basic functionalities provided by mobile app features to keep track of progress such as step counting, recording and using timers emerged to be influential in download behavior. Moreover, 'notifications' also emerged as a significant keyword related to the number of downloads, which highlights the role of feedback mechanisms in download behavior. Thus as another practical implication keywords related to notifications and features to keep track of progress, in addition to providing relevant keywords (see the following two paragraphs summarizing the ANN and RFDT results) in app descriptions will be effective in reaching higher number of downloads according to the findings.

Among analyzed keywords, 'friends' came to light as one of the most influential keywords in the RFDT analysis, indicating that social interactions and ability to share in apps can lead to higher number of downloads. As signified in relevant literature, apps with social networking features tend to perform better in facilitating behavior change [12]. Yet, similar to analogous classification studies, the findings of this study also reveal that only a limited percentage of apps offer sharing and social networking capabilities [56,57]. Similarly, another variable that was found to be an important element is the interactive elements that incorporate 'Users Interact', 'Shares Info', 'Digital Purchase', 'Shares Location', 'Unrestricted Internet' values. The major interactive element with the highest sensitivity, has emerged as 'Users Interact'. This keyword denoting social interaction highlights both the role of social sharing and the interaction between users. Not surprisingly, in online consumer behavior literature, interactivity in

several forms has been found to be a critical element of success in marketing communication, web design, and social media marketing [58,59]. Similar findings of studies in mobile contexts also support the significance of interactivity provided by relevant features in intentions and usage [60–62]. Consequently, providing user interaction has emerged as a viable strategy to influence the total number of downloads and therefore app installs.

Conversely, the last update time of the app has not been found to be a significant factor related to either the number of downloads or user scores. This finding contradicts Krishnan and Selvam's [40] study on diabetes smartphone apps. This may partly be attributed to the differences in the context and the sample. Yet, the 'Android version' variable emerged as a significant factor in the RFDT and the ANN sensitivity analysis. It may be inferred that when the app provides required functionality, works in a wide range of devices (evidenced by the Android version variable) and is free of significant bugs, frequent updates are not vital in the success of the app.

Lastly, the findings suggest that the sponsor/developer of the app is an essential element that is related to the total number of downloads as well as the average user scores. Despite the higher popularity of state-sponsored apps, the user scores are significantly lower in this category. This suggests that more effort and resources should be devoted to the development and improvement of state-sponsored apps that can potentially reach a considerably higher number of people than other sponsor types. In this way, a higher user satisfaction may be established that will most likely turn into a higher usage frequency (i.e. behavior loyalty) and better health outcomes for the general population.

In line with the discussion carried out, the following five practical implications and strategies applicable to a wide range of m-health apps, can be put forward as the major outcomes of the present study:

- Mobile apps should be offered free of cost with in-app purchase options when possible.

- Interactions between users (e.g., social networking and sharing options) should be available among features when relevant.

- Videos should be provided in app description pages.

- Information on privacy policy should be presented in the description section.

- Notifications and similar feedback mechanisms should be specified among app features.

## Conclusion

This study reviewed mobile health apps using content analysis, ANOVA and machine learning algorithms and contributes to the current knowledge on m-health application use in several ways. First, content analysis through manual coding of data yielded added value by providing classifications, keywords and factors that influence mobile health app download behavior and user scores. Second, there is no directly comparable study of this scale carried out regarding mobile health applications specifically in emerging countries such as Turkey. Thus, this study provides guidance to researchers, professionals and policymakers in similar nations as well. Given that the app data is publicly available on marketplaces, similar studies may be carried out to test for adaptability of findings to other contexts.

Despite the effort put into the study, it has a number of limitations. First, this study relied on data provided on app store pages (e.g., descriptions) for categorization and analysis. This created potential discrepancies between the m-health app description and actual features, hence functionality provided to users (i.e. under reporting or over reporting) may have

influenced user scores. This highlights a future research direction that can overcome such discrepancies. The researchers may analyze the mobile applications by personally installing and using them then scoring each app in various dimensions (e.g., features, ease of use, interactivity provided etc.) [63]. Furthermore, the number of app downloads may not be directly indicative of adoption and regular usage behavior as an initial download represents a trial. This disparity points to a further research path that can be focused on. Researchers may enroll users and follow their usage behavior throughout a set time frame via custom mobile apps or personal diaries. By using this tactic, repeat use and adoption behavior may be observed. Furthermore, deeper insights that can complement cross-sectional studies such as the present one may also be obtained. Second, the sample utilized in this study covered only a fraction of all available apps as there are estimated to be more than hundreds of thousands of them. Manual coding of such data is unfortunately not feasible and different approaches are needed to collect and recode such amounts of data. In addition, there is no exhaustive list of apps available to users/researchers in app marketplaces rendering random sampling impractical and forcing researchers to use non-random sampling methods. This leads to a viable research direction, such as carrying out research in collaboration with mobile app marketplace sponsors (e.g., Apple). More generalized findings may be obtained in this way as the researchers can access an exhaustive list of apps, which allows for better sampling.

## Acknowledgments

The authors thank Zehra Nur Canbolat and Omer Berkay Aytac for their assistance with data extraction.

## Author Contributions

**Conceptualization:** Gokhan Aydin.

**Data curation:** Gokhan Silahtaroglu.

**Formal analysis:** Gokhan Silahtaroglu.

**Funding acquisition:** Gokhan Aydin.

**Investigation:** Gokhan Aydin.

**Methodology:** Gokhan Aydin.

**Project administration:** Gokhan Aydin.

**Resources:** Gokhan Silahtaroglu.

**Software:** Gokhan Silahtaroglu.

**Validation:** Gokhan Aydin, Gokhan Silahtaroglu.

**Writing – original draft:** Gokhan Aydin.

**Writing – review & editing:** Gokhan Silahtaroglu.

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
