## [Decision Letter · Decision Letter 0]

8 Sep 2020

PONE-D-20-12673

Insights into mobile health application market via a content analysis of marketplace data with machine learning

PLOS ONE

Dear Dr. Aydin,

Thank you for submitting your manuscript to PLOS ONE. After careful consideration, we feel that it has merit but does not fully meet PLOS ONE’s publication criteria as it currently stands. Therefore, we invite you to submit a revised version of the manuscript that addresses the points raised during the review process.

We look forward to receiving your revised manuscript.

Kind regards,

Farrukh Aslam Khan

Academic Editor

PLOS ONE

Journal Requirements:

2.  Please can you clarify in your methods section whether the two coders provided informed consent to participate in this study."

3. Thank you for including your ethics statement: 

"Istanbul Medipol University Ethical Committee of Non-invasive Clinical Trials. No: 10840098-604.01.01-E.8356".   

Please amend your current ethics statement to confirm that your named institutional review board or ethics committee specifically approved this study.

Once you have amended this statement in the Methods section of the manuscript, please add the same text to the “Ethics Statement” field of the submission form (via “Edit Submission”).

Reviewers' comments:

Reviewer's Responses to Questions

**Comments to the Author**

1. Is the manuscript technically sound, and do the data support the conclusions?

Reviewer #1: Partly

Reviewer #2: No

2. Has the statistical analysis been performed appropriately and rigorously? 

Reviewer #1: No

Reviewer #2: No

3. Have the authors made all data underlying the findings in their manuscript fully available?

Reviewer #1: No

Reviewer #2: Yes

4. Is the manuscript presented in an intelligible fashion and written in standard English?

Reviewer #1: No

Reviewer #2: Yes

5. Review Comments to the Author

Reviewer #1: This manuscript presents an investigation on the current landscape of smartphone apps targeting

to improve and sustain health and well-being. This study contributes to the current knowledge on m-health application use by reviewing the mobile health applications using content analysis and machine learning algorithms. Although the approach is good, however it require major improvements in order to be considered for acceptance.

1. Overall presentation and structure of the paper is not good, Please consider the following comments:

a. Proper alignment (on right margin) is required.

b. The manuscript has many uneven-sized paragraphs, some are very small and other are too long. Please be consistent in that and make apparently same size of paragraphs, while not disturbing the flow of concepts.

c. On page 4. Line no. 14, 16, 18, 20, 22, 24, and page 5, line 1:- Please start sentence with the capital latter for each starting word.

d. Paper has many punctuation, typo, and grammar mistakes, please proofread it carefully.

e. Conclusion must be revised. Please write a concise conclusion. Also add future directions. The shortcoming and other relevant details can be added into a separate section of discussion with more details.

2. Page 9, Line no. 17 and Page 11, line 3 to 11 :- Please provide references to justify your statements.

3. Figures are not clear, Please improve image quality and its description in the text must be detailed. Figure 2 is not acceptable, please redraw it.

4. Please give more details to improve proposed methodology for reader’s understanding. You may add a proposed Algorithm/ flow chart for the ease of the readers.

5. In table 2, column 3, why there is an extra sub-column. In table 3, column 3. Please use 0 (zero) before decimal point, for example, instead of writing “.6125” please write as “0.6125” throughout for readability.

6. Please add a table for different machine learning algorithm output/result comparison in data analysis and result section.

7. Overall, all information (apps data, algorithms, results, etc.) are in textual format, Please rewrite using technical details.

8. Please make graphs to augment the result analysis.

9. References are too old, please add some latest references.

Reviewer #2: The manuscript presents the analysis of mobile health application market and use machine learning models to predict the download behavior of these apps. Following are points which needs to be considered:

1. The machine learning models used in the study are supervised learning algorithms. How the authors labeled the data, for the models?

2. What kind of metrics/ qualitative parameters have been extracted from the raw data to feed into the machine learning model for prediction?

3. How the authors choose the best factors that influence or effect the download behavior? What was the criteria for choosing the factors?

4. The results and analysis section need some more details.

5. English language needs proper revisions

6. PLOS authors have the option to publish the peer review history of their article (what does this mean?). If published, this will include your full peer review and any attached files.

Reviewer #1: No

Reviewer #2: No

---

## [Author Response · Author response to Decision Letter 0]

6 Oct 2020

Reviewer #1: This manuscript presents an investigation on the current landscape of smartphone apps targeting to improve and sustain health and well-being. This study contributes to the current knowledge on m-health application use by reviewing the mobile health applications using content analysis and machine learning algorithms. Although the approach is good, however it require major improvements in order to be considered for acceptance.

1. Overall presentation and structure of the paper is not good, Please consider the following comments:

a. Proper alignment (on right margin) is required. 

Author(s): Thank you for highlighting this issue, the text is properly aligned. 

b. The manuscript has many uneven-sized paragraphs, some are very small and other are too long. Please be consistent in that and make apparently same size of paragraphs, while not disturbing the flow of concepts.

Author(s): Several improvements were made throughout the text to improve the readability, formatting and paragraph sizes. 

c. On page 4. Line no. 14, 16, 18, 20, 22, 24, and page 5, line 1:- Please start sentence with the capital latter for each starting word.

Author(s): Thank you, the problems are corrected. 

d. Paper has many punctuation, typo, and grammar mistakes, please proofread it carefully.

Author(s): The paper was proofread by a native speaker to rectify typos and grammar mistakes. 

e. Conclusion must be revised. Please write a concise conclusion. Also add future directions. The shortcoming and other relevant details can be added into a separate section of discussion with more details.

Author(s): We have revised the Conclusion and removed the irrelevant section to incorporate related future research directions. 

2. Page 9, Line no. 17 and Page 11, line 3 to 11 :- Please provide references to justify your statements.

Author(s): Reference(s) has been amended to the related section(s). 

3. Figures are not clear, Please improve image quality and its description in the text must be detailed. Figure 2 is not acceptable, please redraw it.

Author(s): The figure has been completely changed and redrawn. 

4. Please give more details to improve proposed methodology for reader’s understanding. You may add a proposed Algorithm/ flow chart for the ease of the readers.

Author(s) An overall model has been added as Figure 2 to be clearer about the model used in the analysis. Moreover, more details on the analysis method and the algorithms are amended into the paper. 

5. In table 2, column 3, why there is an extra sub-column. In table 3, column 3. Please use 0 (zero) before decimal point, for example, instead of writing “.6125” please write as “0.6125” throughout for readability.

Author(s): The extra column has been removed and 0 was amended before decimal point(s). 

6. Please add a table for different machine learning algorithm output/result comparison in data analysis and result section.

Authors: The results and performances of each algorithm are presented on Table 4. Yet, a figure has been amended into the relevant section for better visual communication. After the first run of analysis, an extra ANN analysis was performed, and the results are displayed on Table 5. This part has been made clearer in the manuscript. Thank you for the warning. 

7. Overall, all information (apps data, algorithms, results, etc.) are in textual format, Please rewrite using technical details.

Author(s): Formulas and figures were amended into the paper to make it easier to understand the logic and mathematical foundations of each algorithm applied. 

8. Please make graphs to augment the result analysis.

Author(s): A pie chart has been added in the discussion section to make final results clearer. Moreover, a chart depicting the performance of each machine learning algorithm was amended to the text

9. References are too old, please add some latest references.

Author(s): Six new studies published in the last two years are amended into the text. 

● Agnihothri, S., Cui, L., Delasay, M., & Rajan, B. (2020). The value of mHealth for managing chronic conditions. Health Care Management Science, 23, 185–202. https://doi.org/10.1007/s10729-018-9458-2

● Bettiga, D., Lamberti, L., & Lettieri, E. (2020). Individuals’ adoption of smart technologies for preventive health care: a structural equation modeling approach. Health Care Management Science, 23, 203–214. https://doi.org/10.1007/s10729-019-09468-2

● Davalbhakta, S., Advani, S., Kumar, S., Agarwal, V., Bhoyar, S., Fedirko, E., Misra, D. P., Goel, A., Gupta, L., & Agarwal, V. (2020). A Systematic Review of Smartphone Applications Available for Corona Virus Disease 2019 (COVID19) and the Assessment of their Quality Using the Mobile Application Rating Scale (MARS). Journal of Medical Systems, 44(9). https://doi.org/10.1007/s10916-020-01633-3

● Levine, D. M., Co, Z., Newmark, L. P., Groisser, A. R., Holmgren, A. J., Haas, J. S., & Bates, D. W. (2020). Design and testing of a mobile health application rating tool. Npj Digital Medicine, 3(1). https://doi.org/10.1038/s41746-020-0268-9

● Rowland, S. P., Fitzgerald, J. E., Holme, T., Powell, J., & McGregor, A. (2020). What is the clinical value of mHealth for patients? Npj Digital Medicine, 3(1). https://doi.org/10.1038/s41746-019-0206-x

● Zhang, Y., Liu, C., Luo, S., Xie, Y., Liu, F., Li, X., & Zhou, Z. (2019). Factors influencing patients’ intention to use diabetes management apps based on an extended unified theory of acceptance and use of technology model: Web-based survey. Journal of Medical Internet Research, 21(8), 1–17. https://doi.org/10.2196/15023

Reviewer #2: The manuscript presents the analysis of mobile health application market and use machine learning models to predict the download behavior of these apps. Following are points which needs to be considered:

1. The machine learning models used in the study are supervised learning algorithms. How the authors labeled the data, for the models?

Author(s): For classification the total number of downloads has been used as the class variable. This part has been made clearer in the manuscript. Thank you for highlighting this issue. 

2. What kind of metrics/ qualitative parameters have been extracted from the raw data to feed into the machine learning model for prediction?

Author(s): A detailed table, Table 1 Code Sheet, presents the metrics/ parameters that have been extracted and fed into the machine learning models. This has been made clearer in the manuscript, thank you for pointing this out. 

3. How the authors choose the best factors that influence or effect the download behavior? What was the criteria for choosing the factors?

Author(s): All the relevant data available on marketplaces regarding each app was put into consideration. This study focuses on the features and keywords that can be extracted from the textual data available in mobile app descriptions as well. Consequently, studies on mobile app acceptance were used as guidelines yet the machine learning algorithms provided distinct keywords related to the download behavior and user scores that were not known to the researchers beforehand. – 

4. The results and analysis section need some more details.

Author(s): A new table and a chart has been amended into the results section. Moreover, several formulas and a chart has been added to the analysis section to make the algorithms and the analysis methodology easier to understand. The results section is also expanded using new graphs and several amendments to the text. 

5. English language needs proper revisions

Author(s): The paper was proofread by a native speaker to rectify typos and grammar mistakes.

---

## [Decision Letter · Decision Letter 1]

11 Nov 2020

PONE-D-20-12673R1

Insights into mobile health application market via a content analysis of marketplace data with machine learning

PLOS ONE

Dear Dr. Aydin,

Thank you for submitting your manuscript to PLOS ONE. After careful consideration, we feel that it has merit but does not fully meet PLOS ONE’s publication criteria as it currently stands. Therefore, we invite you to submit a revised version of the manuscript that addresses the points raised during the review process.

We look forward to receiving your revised manuscript.

Kind regards,

Farrukh Aslam Khan

Academic Editor

PLOS ONE

Additional Editor Comments (if provided):

Please carefully check the paper for English language mistakes including punctuation, typos, etc. I would recommend getting the paper checked by a native speaker.

Reviewers' comments:

Reviewer's Responses to Questions

**Comments to the Author**

1. If the authors have adequately addressed your comments raised in a previous round of review and you feel that this manuscript is now acceptable for publication, you may indicate that here to bypass the “Comments to the Author” section, enter your conflict of interest statement in the “Confidential to Editor” section, and submit your "Accept" recommendation.

Reviewer #1: (No Response)

Reviewer #2: All comments have been addressed

2. Is the manuscript technically sound, and do the data support the conclusions?

Reviewer #1: Yes

Reviewer #2: Yes

3. Has the statistical analysis been performed appropriately and rigorously? 

Reviewer #1: Yes

Reviewer #2: Yes

4. Have the authors made all data underlying the findings in their manuscript fully available?

Reviewer #1: Yes

Reviewer #2: Yes

5. Is the manuscript presented in an intelligible fashion and written in standard English?

Reviewer #1: Yes

Reviewer #2: Yes

6. Review Comments to the Author

Reviewer #1: I am Happy that the comments are addressed carefully, however, still there are few minor punctuation mistakes that me be addressed:

1. Please use a comma "," after e.g. as e.g., throughout.

Reviewer #2: Most of the revision prompted by the comments has been incorporated and no further explanation is required.

7. PLOS authors have the option to publish the peer review history of their article (what does this mean?). If published, this will include your full peer review and any attached files.

Reviewer #1: **Yes: **Noshina Tariq

Reviewer #2: No

---

## [Author Response · Author response to Decision Letter 1]

13 Nov 2020

Reviewer #1: I am Happy that the comments are addressed carefully, however, still there are few minor punctuation mistakes that me be addressed:

1. Please use a comma "," after e.g. as e.g., throughout.

Authors: The authors thank the reviewer the comments and suggestions that definitely helped in improving the manuscript. This minor issue has been settled and a comma was added after `e.g.` throughout the manuscript.

Reviewer #2: Most of the revision prompted by the comments has been incorporated and no further explanation is required.

Authors: The authors thank the reviewer for the valuable input that helped us in improving the manuscript.

---

## [Decision Letter · Decision Letter 2]

8 Dec 2020

Insights into mobile health application market via a content analysis of marketplace data with machine learning

PONE-D-20-12673R2

Dear Dr. Aydin,

We’re pleased to inform you that your manuscript has been judged scientifically suitable for publication and will be formally accepted for publication once it meets all outstanding technical requirements.

Kind regards,

Farrukh Aslam Khan

Academic Editor

PLOS ONE

Additional Editor Comments (optional):

The authors have addressed all the reviewers’ comments. The paper is in good shape now.

Reviewers' comments:

Reviewer's Responses to Questions

**Comments to the Author**

1. If the authors have adequately addressed your comments raised in a previous round of review and you feel that this manuscript is now acceptable for publication, you may indicate that here to bypass the “Comments to the Author” section, enter your conflict of interest statement in the “Confidential to Editor” section, and submit your "Accept" recommendation.

Reviewer #1: All comments have been addressed

2. Is the manuscript technically sound, and do the data support the conclusions?

Reviewer #1: Yes

3. Has the statistical analysis been performed appropriately and rigorously? 

Reviewer #1: Yes

4. Have the authors made all data underlying the findings in their manuscript fully available?

Reviewer #1: Yes

5. Is the manuscript presented in an intelligible fashion and written in standard English?

Reviewer #1: Yes

6. Review Comments to the Author

Reviewer #1: (No Response)

7. PLOS authors have the option to publish the peer review history of their article (what does this mean?). If published, this will include your full peer review and any attached files.

Reviewer #1: **Yes: **Noshina Tariq

---

## [Editor Report · Acceptance letter]

15 Dec 2020

PONE-D-20-12673R2 

Insights into mobile health application market via a content analysis of marketplace data with machine learning 

Dear Dr. Aydin:

I'm pleased to inform you that your manuscript has been deemed suitable for publication in PLOS ONE. Congratulations! Your manuscript is now with our production department. 

Kind regards, 

on behalf of

Dr. Farrukh Aslam Khan 

Academic Editor

PLOS ONE